# Compressive Alginate Sponge Derived from Seaweed Biomass Resources for Methylene Blue Removal from Wastewater

**DOI:** 10.3390/polym11060961

**Published:** 2019-06-02

**Authors:** Xiaojun Shen, Panli Huang, Fengfeng Li, Xiluan Wang, Tongqi Yuan, Runcang Sun

**Affiliations:** 1Beijing Key Laboratory of Lignocellulosic Chemistry, Beijing Forestry University, Beijing 100083, China; shenxiaojun@iccas.ac.cn (X.S.); huangpanli@bjfu.edu.cn (P.H.); qhxylifeng@163.com (F.L.); rcsun3@bjfu.edu.cn (R.S.); 2Beijing National Laboratory for Molecular Sciences, Key Laboratory of Colloid and Interface and Thermodynamics, Institute of Chemistry, Chinese Academy of Sciences, Beijing 100190, China; 3Center for Lignocellulose Science and Engineering, and Liaoning Key Laboratory Pulp and Paper Engineering, Dalian Polytechnic University, Dalian 116034, China

**Keywords:** alginate sponge, two-step lyophilization, methylene blue, adsorption capacity, biomass resources

## Abstract

Low cost fabrication of water treatment polymer materials directly from biomass resources is urgently needed in recent days. Herein, a compressive alginate sponge (AS) is prepared from seaweed biomass resources through a green two-step lyophilization method. This material is much different from conventional oven-, air-, vacuum-dried alginate-based adsorbents, which show limitations of shrinkage, rigidness, tight nonporous structure and restricted ions diffusion, hindering its practical applications, and was used to efficiently remove methylene blue (MB), a main colorful contaminant in dye manufacturing, from wastewater. The batch adsorption studies are carried out to determine the impact of pH, contact time and concentration of dye on the adsorption process. The maximum adsorption capacity can be obtained at 1279 mg g^−1^, and the shape-moldable AS can be facilely utilized as a fixed-bed absorption column, providing an efficient approach for continuous removal of MB within a short time. It is also important that such a compressive AS can be regenerated by a simple squeezing method while retaining about 70% capacity for more than ten cycles, which is convenient to be reused in practical water treatment. Compressive AS demonstrates its merits of high capability, large efficiency and easy to recycle as well as low cost resources, indicating widespread potentials for application in dye contaminant control regarding environmental protection.

## 1. Introduction

Recently, industrial developments led to the undesirable environmental problems in the world, especially water pollution [1,2,3]. Many industries like the textile industry produce much wastewater, which contains a number of contaminants, including acidic or caustic dissolved solids, toxic compounds, and any different dyes, many of these dyes are carcinogenic, mutagenic, and teratogenic and also toxic to human beings, fish species, and microorganisms [1,4,5,6,7,8]. Among these contaminants, dyes are common contaminants in water discharged from various industries, such as plastics, textiles, printing, paper and leather industry. Due to its complex aromatic molecular structures, dyes in contaminated water are stable and resistant to biodegradation, which has become a serious environmental problem [9,10,11]. Therefore, the development of industrially applicable and efficient treatments for dye containing effluents is urgently needed [12,13,14]. Technologies such as coagulation/flocculation [15], biological treatment [16], oxidation or ozonation [17], membrane separation [18], electrochemical methodology [19] and adsorption [20] have been extensively applied to remove dye pollutants from wastewater. Among them, adsorption method is quite popular due to its convenience, ease of operation, low energy consumption and availability of various adsorbents [21,22,23,24]. Currently, a number of adsorbent materials, such as activated carbon [25], mesoporous silica [26], hybrid xerogel [27], nanoporous alumina [28], zeolites [29] and carbon-based nanomaterials [13], have demonstrated their capability as an effective adsorbent for dye contaminants. However, their widespread use in water treatment is sometimes restricted due to its high cost and complexity of material preparation.

Recently, the three-dimensional (3D) porous structures ensure a large surface area for efficient dye adsorption, and exhibits the desirable merits of biocompatibility, biodegradability and economic efficiency [30]. Hence, much attention has been paid to developing sponge and porous materials as absorbers and they can achieve oil−water separation via a simple, fast, and effective absorption process [31,32,33,34,35]. Generally, an ideal absorbent material should have high oil absorption capacity, high selectivity, low density, and excellent recyclability, and it should be environmentally friendly. A large variety of low-cost adsorbents including natural species (plant fiber [36], silk cotton [37], leaves [38], etc.), industrial/agricultural wastes or by-product (citrus waste peels [39], bagasse [40], rice husk [41], etc.) and extractive biopolymers (chitosan [42], guar gum [43], alginate [44,45,46,47], etc.) have been explored as they are inexpensive, non-toxic and environmentally friendly. In the past few years, numerous efforts have been focused on the development of more effective and cheaper adsorbents derived from natural biomass resources rather than commercial adsorbents. Alginate, isolated from natural brown seaweeds, is a linear, unbranched and anionic polysaccharide biopolymer composed of β-D-mannuronic acid (M) and α-L-guluronic acid (G) units in the form of homopolymeric (MM- or GG-blocks) and heteropolymeric sequences (MG or GM-blocks) [48,49,50]. Alginate can be facilely converted into a porous sponge monolith forming the so-called “egg-box” conjunct structure [51]. In addition, the presence of negative carboxylate functional groups along polymer chains provides sufficient active sites, which guarantees its high affinity and binding ability for dye species [52,53]. Previous studies have demonstrated that conventional oven-, air-, vacuum-dried alginate-based adsorbents show limitations of shrinkage, rigidness, tight nonporous structure and restricted ions diffusion, hindering its practical applications [54].

In the paper, our aim is the fabrication of sponge-like adsorbing polymers directly from biomass resources, which is consistent with the object of low cost and renewable material design. Herein, we describe a green two-step lyophilization strategy to prepare a compressive alginate sponge (AS) derived from natural seaweed biomass resources with a large surface area, high porosity and controllable morphology. This method has advantages of low cost, simple and eco-friendly approach that does not include toxic or expensive resources, complicated or multistep chemical reactions. In this work, the as-prepared sponge was used to efficiently remove methylene blue (MB), a main colorful contaminant in dye manufacturing, from aqueous solution through the batch and continuous fixed-bed column adsorption. Their kinetics, thermodynamic analysis and adsorption mechanism are investigated. More importantly, the regeneration of the compressive sponge just by squeezing demonstrates its large potential to be used in practical applications for water treatment.

## 2. Materials and Methods

### 2.1. Materials

Waste brown seaweeds were collected from Shandong province, China. It was exhaustively washed with tap water, air-dried, and cut into pieces before use. Sodium carbonate (≥99.8%), calcium chloride (≥96.0%), hydrochloric acid (36.0~38.0%), sodium hydroxide (≥96.0%), methylene blue (≥90%) were purchased from Sinoparm Chemical Reagent Beijing Co., Ltd (Beijing, China).

### 2.2. Extraction of Seaweed

Sodium alginate was extracted from seaweed resources according to our previous work in our lab [43]. Firstly, the pigments of seaweed samples (5 g) on the surface were fixed by soaking in 1 wt % formaldehyde solution (300 mL). Then, the seaweeds were treated with 4 wt % Na_2_CO_3_ (300 mL) at 46 °C for 3 h. After that, the mixture was filtered, neutralized and precipitated by dropping the 10 wt % CaCl_2_ to obtain the calcium alginate. The crude calcium alginate was washed thoroughly with distilled water and then acidified with 5 wt % HCl for 1 h. After that, gel materials of alginic acid were gained by filtering and washed several times with distilled water. The alginic acid was then dispersed in distilled water and converted into sodium alginate by neutralization (pH 6–7) with 1 wt % NaOH. Finally, the sodium alginate was precipitated and purified by ethanol, and collected for further use after freeze-drying. Its weight-average molecular weight (Mw) and M/G (M is β-D-mannuronic acid and G is α-L-guluronic acid) value were 1.51 × 10^6^ g mol^−1^ and 0.86, respectively.

### 2.3. Preparation of AS

The AS was prepared by a green two-step lyophilization method from seaweed biomass resources, which was presented in Scheme 1. 0.2 g sodium alginate powder extracted from seaweed biomass resources was dissolved in 9.8 g distilled water (2 wt %) under vigorous stirring until homogeneous dispersion was formed. Then, the alginate solution was carefully cast into Petri dish plates and freeze-dried at −50 °C for 48 h. After that, the lyophilized sodium alginate sample was rinsed with 5 wt % CaCl_2_ aqueous solution under quiescent conditions. The ionic crosslinking process was maintained for 12 h allowing the Ca^2+^ ions to be fully crosslinked with sodium alginate. Hereafter, the crosslinked alginate sample was soaked with distilled water and washed for several times to remove the excessive salts. After that, the ionic crosslinked sample was freeze-dried at −50 °C for 48 h. The as-obtained AS could be cut and shaped into different geometrical shapes easily by scissors and cutters. For clearly understanding, the involved chemical equations of different steps in Scheme 1 were supported by Appendix A.

### 2.4. Characterizations

The morphology of the AS was examined by the field-emission scanning electron microscope (SEM) (HitachiS-5500, Tokyo, Japan) operating at 10 kV acceleration voltage. All samples were coated with gold prior to acquiring the images. The pore size distributions were calculated by the Mercury Intrusion Porosimetry (Demo AutoPore IV 9500, Micromeritics, Atlanta, GA, USA). The Fourier transform infrared spectroscopy (FT-IR) spectra of the AS was collected by a Thermo Scientific Nicolet iN10 FT-IR Microscope (Thermo Nicolet Corporation, Waltham, MA, USA) equipped with an MCT detector cooled by liquid nitrogen in the reflection mode. The spectra were recorded in the range from 4000 to 670 cm^−1^ with 64 scans at a 4 cm^−1^ resolution. The crystalline structure measurement was performed on an XRD-7000 X-ray diffractometer (Shimadzu, Kyoto, Japan) with a Cu Kα radiation source (λ = 0.154 nm) at 40 kV and 30 mA. The X-ray photoelectron spectra (XPS) were determined on the photoelectron spectrometer (ESCALAB 250Xi, Thermo Fisher, Waltham, MA, USA) using an Al Kα (1486.6 eV) radiation. Mechanical properties were carried out on a UTM6503 universal tensile Machine (Shenzhen Suns Technology stock CO. LTD. Shenzhen, China) equipped with a 100 N load cell at the rate of 10 mm min^−1^. The thermal behaviour was studied by thermogravimetric analyses (TGA) using a DTG-60 simultaneous thermal analyzer (Shimadzu, Tokyo, Japan) under nitrogen atmosphere. 3–6 mg samples were loaded in an alumina crucible and heated from 40 °C to 900 °C at the rate of 10 °C min^−1^.

### 2.5. Adsorption Measurement

The dye adsorption capability for AS was investigated at room temperature with methylene blue (MB). The MB solution was prepared by dissolving the MB powder in deionized water. Then, it was diluted with deionized water into a desired concentration varied between 50–1800 mg L^−1^. Typical batch adsorption tests were accomplished by suspending 20 mg AS in 50 mL MB aqueous solution with different concentrations at an initial natural pH value in a flask. Afterwards, the flask was continuously shaken in a rotary shaker with 100 rpm for a period time to reach the adsorption equilibrium. In the study of the pH effect, the required pH value of the solution was adjusted with 0.1 mol L^−1^ HCl or NaOH aqueous solutions. After the adsorption equilibrium, the residual concentrations of the MB supernatants were evaluated by an UV-vis spectrometer at the absorbance of 664 nm. The amount of MB adsorbed by AS was calculated as following equation:(1)$$Q_{e}\;=\;\frac{\left(C_{0}\;{-\;C}_{e}\right)V}{m}$$
where *Q*_e_ (mg g^−1^) represents the adsorption capacity at equilibrium, *C*_0_ (mg L^−1^) and *C*_e_ (mg L^−1^) are the MB concentrations at the initial and equilibrium, respectively. *V* is the volume of the solution (mL), and *m* is the mass of dry adsorbent (mg).

The continuous fixed-bed column adsorption experiments were conducted in a U-tube glass column with an inner diameter of 0.8 cm and volume capacity of 15 mL. The AS was cut with the desired column (diameter: 0.8 cm, height: 0.45 cm, weight: 7.8 mg) and then packed into the glass column tightly. Two layers of adsorbent cotton and bugles (diameter 400–800 μm) were embedded on the top and bottom of the packed column. 50 mL MB solution with an initial concentration of 1000 mg L^−1^ was pumped through the column in a down-flow mode using a peristaltic pump with the rate of 10 mL min^−1^ at room temperature.

## 3. Results and Discussions

### 3.1. Material Properties

The as-obtained sponge was elastic enough to be easily handled and cut into desired shapes, such as rods, cylinders, papers or cubes to satisfy various absorption requirements (Figure 1a). Moreover, the AS has an ideal density (34.6 mg cm^−3^) compared with conventional compacted alginate-based materials [54], which could effortlessly stand on a grass leaf (Figure 1b). The strength and lightweight of AS were closely related to its three-dimensional (3D) porous framework with abundant pores as shown in Figure 1c,d. As a result, the interior sponge possessed a wide range of pore sizes from nanometers to hundreds of micrometers (mostly less than 100 μm) calculated by the mercury intrusion porosimetry method (Appendix A). The FT-IR analysis and XRD pattern revealed that the carboxyl groups penetrated in the AS network and the AS has none of the distinct crystalline formation (Figure 1e,f). Through the photoelectron lines of the wide-scan XPS spectrum (Appendix A), the presence of carbon (57.72% area), oxygen (39.03% area) and calcium (3.25% area) atoms in the AS were confirmed. Further investigation of atomic binding states on the surface of the AS was conducted by XPS analysis (Figure 1g,h).

### 3.2. Absorption Properties

The 3D multiporosity and large active surfaces of AS make it an ideal candidate for the removal of methylene blue (MB) dye from wastewater. It should be noted that the initial pH value plays a significant role in the adsorption performances. It determines the adsorption reactions between MB and the adsorbent surface, attributing to the ionization in the solution and charge distribution on the adsorbent surface. In order to optimize the pH value to obtain the maximum adsorption capacity, batch adsorption experiments were conducted at pH values ranging from 2.0 to 10.0 with an initial MB concentration of 1000 mg L^−1^ for a contact time of 12 h. As shown in Figure 2a, the adsorption capacity was strongly related to the initial pH value. The lowest valve of *Q*_e_ (188 mg g^−1^) was found at the pH value of 2.0. It can be speculated that surfaces of AS were highly protonated due to the high concentration of H^+^. Thus, the –COO^−^ active sites for adsorption of MB were occupied and those surface negative charges that were reduced, weaken the electrostatic attraction forcers between surfaces of AS and MB. Resultantly, the adsorption capacity is decreased with the decrease of pH from 4 to 2. However, when the pH value was higher than 4.0 (above the pKa value of MB), the surface charges of AS became more negative which was confirmed by Zeta potential data, promoting the electrostatic interaction between surface active sites and adsorbate, improving the adsorption capacity for MB. Additionally, the presence of excess of H^+^ ions at low pH will compete with the MB cations for adsorption sites, resulting in a decrease of adsorption capacity, while more negatively charged surfaces of adsorbents become available with increased pH value [55,56]. This efficient adsorption behavior of the AS at pH = 4–10 demonstrates that it is potentially applicable in a relatively wide pH range. Nonetheless, considering the instability of calcium alginate in the strong alkaline condition, all the following experiments were conducted at pH = 4.

To evaluate the adsorption capacity of AS for MB, adsorption isotherms with different initial MB concentrations (5–1800 mg L^−1^) were measured at room temperature of 25 °C with a contact time of 12 h. As shown in Figure 2b, it could be distinctly observed that the value of adsorption capacity increased with the increase of initial MB concentrations, until trended toward the equilibrium state. This can explain that the higher initial concentration generated a stronger driving force resulting from the concentration gradient, in favor of a mass transfer for MB from aqueous solution to adsorbent. With the initial concentration up to 1000 mg L^−1^, the active adsorption sites of the adsorbent were almost occupied by MB molecules, thus the adsorption capacity attained at the maximum value and kept constant even at higher initial concentration of MB [57]. The equilibrium adsorption capacity could finally reach at 1279 mg g^−1^ with the initial MB concentration of 1800 mg L^−1^. In addition, we also calculated the removal ratio of the sponge at different initial concentrations of MB. A higher removal ratio about 90% was obtained at the relatively low initial concentration. Even at low concentrations of 5 and 10 mg L^−1^, the AS also has high sensitivity for efficient MB removal, which is important for adsorbents used in the water treatment. A clear comparison of the removal effect could be presented by the images before and after batch adsorption (Appendix A). In additional, the AS before and after adsorbing methylene blue was characterized by the FT-IR spectra. As shown in Appendix A, the infrared characteristic peaks (1607 and 1422 cm^−1^) of the C=O group shifted to low infrared absorption peak, which suggested that the internal porous structure of AS was favorable to the electrostatic interaction between the polymer carboxylic functional groups and cationic groups of the MB.

The adsorption isotherm is essential for the adsorption to further explore its adsorption mechanism. In this study, the Langmuir and Freundlich isotherm models were used to fit the adsorption data, respectively, based on the adsorption isotherm of MB (Figure 2c). The Langmuir isotherm model assumes its monolayer adsorption over a homogeneous adsorbent surface with adsorption sites that are identical and energetically equivalent. Therefore, it can achieve at a saturate state without further adsorption activities. On the other hand, the Freundlich isotherm model is an empirical equation, which is employed to describe a multilayer adsorption onto heterogeneous surfaces with undefined sites. The two isotherm models can be described by two equations as follows:(2)$$\frac{C_{e}}{Q_{e}}\;=\;\frac{C_{e}}{Q_{m}}\;+\;\frac{1}{K_{L}Q_{m}}\;$$
(3)$${\mathrm{logQ}}_{e}{=\;\mathrm{logK}}_{F}\;+\;\frac{1}{n}{\mathrm{logC}}_{e}$$
where *C*_e_ is the final equilibrium concentration (mg L^−1^), *Q*_e_ is the adsorption capacity at equilibrium (mg g^−1^), *Q*_m_ is the maximum adsorption capacity (mg g^−1^), *K*_L_ is the Langmuir constant (L mg^−1^), *K*_F_ is a constant related to the adsorption capacity (mg g^−1^) (L mg^−1^) ^1/n^ and *n* is an empirical parameter related to the adsorption intensity.

The linearized Langmuir and Freundlich adsorption isotherm plots were drawn as Figure 2d and their parameters were calculated from the adsorption isotherms listed in Table 1. The higher correlation coefficient (*R*^2^ = 0.9991) of the Langmuir isotherm model indicated that it was better than the Freundlich isotherm model (*R*^2^ = 0.8515) in simulating our experiment data. The fitting result presented that the adsorption behavior occurred on homogeneous surfaces of AS through a monolayer manner. Moreover, according to the Langmuir fitting result, the maximum adsorption capacity for MB of AS was 1310 mg g^−1^ in high concordance with the experimental equilibrium value (1279 mg g^−1^), which is comparable with alginate based or other reported adsorbents for the MB adsorption listed in Appendix A. Importantly, the synthesis of the AS adsorbent is quite green, economic and might be applied for large-scale production.

The adsorption kinetics was examined by measuring the adsorption capacity at initial MB concentrations of 500 and 1000 mg L^−1^, respectively. The samples were collected at different time intervals up to 12 h. Figure 3a shows the effect of contact time on the adsorption capacity of AS for MB with different initial concentrations at an ambient temperature. It demonstrated that the adsorption capacity increased rapidly at the initial adsorption stage, then increased with the contact time at a relatively slow rate, and reached up to the equilibrium within 390 min. The adsorption capacity increased with higher initial concentrations, since the surface coverage was relatively low in the early stage, thus dye molecules could occupy the vacant adsorption sites rapidly. Once the adsorption process approached at the equilibrium, there would be less vacant active sites. Furthermore, the adsorbed MB molecules on the surface of AS could repel free MB molecules via the electrostatic repulsion.

To investigate the kinetic mechanism, the pseudo-second-order equation was used to describe the adsorption process. The time-dependent adsorption process was analysed using the pseudo-first-order and pseudo-second-order kinetic models (Figure 3b) according to the following equations:(4)$$\log\left(Q_{1e}-Q_{t}\right)\;{=\;\mathrm{logQ}}_{1e}-\frac{k_{1}t}{2.303}\;$$
(5)$$\frac{t}{Q_{t}}\;=\;\frac{1}{k_{2}Q_{2}{}^{2\;}}+\;\frac{t}{Q_{2e}}\;$$
where *k*_1_ (min^−1^) is the pseudo-first-order rate constant; *k*_2_ [g (mg·min)^−1^] is the pseudo-second-order kinetic rate constant; *Q*_e_ and *Q*_t_ are the amounts (mg g^−1^) of adsorption at the equilibrium and at contact time referred as *t* (min). The pseudo-first-order assumes that the adsorption rate linear declines with the increase of the removal efficiency. The pseudo-second-order holds the opinion that the rate-controlling step of adsorption is the interaction between the adsorbent and adsorbate, such as ion sharing and transferring [19].

The kinetic model parameters were obtained by the regression analysis of the experimental data (Table 2). It can be observed that the *R^2^* values for the pseudo-first order kinetic model was lower than that for the pseudo-second-order kinetic model. The pseudo-second-order kinetic model (*R*^2^ = 0.9988) provided a better correlation in contrast to the pseudo-first-order kinetic model (*R*^2^ = 0.9890) for MB adsorption. Additionally, the values of *Q*_1e_ (694 mg g^−1^) failed to predicate the values of *Q*_exp_, while the values of *Q*_2e_ (943 mg g^−1^) is pretty close to *Q*_exp_ (845 mg g^−1^), confirming that the kinetics of MB adsorption was identical when described by the pseudo-second-order kinetic model rather than the pseudo-first-order model. It means that the overall adsorption rate seems to be controlled by the chemical interaction through covalent forces during the process of exchanging electrons between adsorbent and adsorbate.

Based on previous reported literature, the porous structure of AS favors the electrostatic interaction between the polymer carboxylic functional groups and cationic groups of the MB. A possible adsorption mechanism for MB adsorption was shown in Scheme 2. According to the ‘‘egg-box’’ model of gelation mechanism, the carboxyl functional groups of the *α*-L-guluronic acid in sodium alginate could form ionic bonds with divalent Ca^2+^ ions, leading to the transformation of the alginate solution into a stable sponge and forming abundant holes, which could trap MB molecules by the carboxyl functional groups in the network of AS. In addition, the hydroxyl groups of AS could also stabilize the MB–alginate complex.

### 3.3. Compressive Properties and Recyclability

The performance of adsorbents in the continuous dyes adsorption process is an important factor in assessing the feasibility of adsorbents in real-time practical applications. Fortunately, the free-standing and elastic AS can be easily handled, superior to the other suspended adsorbents. The compressive properties of AS were examined by mechanical tests which were repeated within compress-release cycles. The AS sample exhibited excellent elastic resilience under compression, it could recover to its original morphology after removal of the external load (Figure 4a). The cyclic compression stress-strain (σ-ε) curves at a maximum strain of 30%, 50%, and 70% were shown in Figure 4b. In the first circle, the loading curve exhibited three distinct stages: (i) A linear-elastic regime during which the stress increases linearly at ε < 20%, the sponge started to bend and the structure of the interior pore began to change; (ii) a plateau region at 20% < ε < 60%, the sponge was compressed to a certain extent and the increase of stress was slow; (iii) a steep slope region at ε > 60% with rapidly rising stress because of the densification of the sponge, where the stress increases linearly again. Even under a maximum strain of 70%, AS can still recover its original elasticity after removal of the external load. Moreover, the maximum value of compression stress increased from 17.7 to 96.7 KPa with compression strain increased from 30% to 70%. Cyclic compression tests were also conducted as shown in Figure 4c. With the increase of the compression cycles, the hysteresis loop for the multicycle curve shrinks compared to the first circle, while after the 5th cycle almost coincided, indicating that the plastic deformation increased but gradually become stabilized. However, the maximum stress remains unchanged, suggesting that AS can retain its compressive strength for 10 cycles with the maximum strain of 50%. These phenomena can be explained by the mechanical elasticity of the pore walls constructed during the two-step lyophilization process.

Thus, we fabricated a mechanically strong and free-standing AS sample (height 1.2 cm, volume capacity 15 mL, internal diameter 0.8 cm) into a flexible column device for fast removing MB from water (Figure 5a). Different from the traditional shaking adsorption in the vessel, the fixed-bed column adsorption contains dynamic behaviors allowing continuous operation. The dynamic adsorption process under constant flow provided a more efficient pathway to utilize the internal multipores in the network of AS, thus improving the adsorption efficiency in quite a short time. The continuous adsorption experiments were undertaken with different contact times treating 50 mL MB aqueous solution. The initial concentration of MB was set as 1000 mg L^−1^. The adsorption capacities along with the different contact time were presented in Figure 5b. It can be concluded that the adsorption capacity increased with the increasing of the contact time, and reached up to a maximum capacity of 1100 mg g^−1^ within 50 min. Further prolonging the contact time could not enhance the adsorption capacity. This is mainly due to the fact that the interaction surface of AS can be utilized sufficiently in the first 50 min, and the further increase of the contact time cannot improve the efficient interaction surface area. The fixed-bed column adsorption demonstrates a large potential for practical application owing to their higher efficiency as compared with traditional shaking adsorption conditions (1279 mg g^−1^ in 720 min). The column-packed adsorbent could treat quantities of MB contained waste one time, which was considered to be an economic waste treatment.

The recyclability is also important in pollution control applications. A simple squeezing method was applied facilely to recycle the adsorbed AS and harvest the pollutants due to its compressive properties. The recyclability test was investigated through fixed-bed column adsorption. As shown in Figure 5c, the adsorption capacity for MB was maintained for about 70% after 10 cycles with 861 mg g^−1^, exhibiting good recyclability upon the easy squeezing method. The main reason for the residual MB during the recycled use may be attributed to the stability of partial trapped MB molecules complexed within the network of AS chains. For further thoroughly removal of MB during the cyclic tests, chemical cleaning or heat treatment might be efficient.

Thermal stability is a key criterion to characterize the temperature limit for the environment of the absorbent during the procedure of adsorption. As shown in Appendix A, the first weight loss (about 10%) of AS in the temperature range from 40 to 190 °C was corresponded to the evaporation of free water and water-linked hydrogen bonds [52,53]. The second mass loss of almost 40% at 190–400 °C was mainly due to the thermal destroy of glycosidic bonds for alginate polar interactions with the carboxylate groups. The final step of mass loss (400–900 °C) might be attributed to the carbonate formation under high temperature [54]. The final thermal degradation of alginate could be the result of forming CaCO_3_.

## 4. Conclusions

A compressive AS was prepared from seaweed biomass resources via a green two-step lyophilization method, which was used to remove MB from wastewater for pollutant control. Due to the 3D multiporous structure, large specific surface area and sufficient active sites, the as-prepared AS exhibited high adsorption capacity (1279 mg g^−1^) for MB removal, which is much different from the conventional oven-, air-, vacuum-dried alginate-based adsorbents, which show limitations of shrinkage, rigidness, tight nonporous structure and restricted ions diffusion, hindering its practical applications. Their kinetics, thermodynamic analysis and adsorption mechanism are investigated. Importantly, the freestanding and elastic AS was facilitated to assemble into a column-packed device for fixed-bed continuous wastewater treatment, indicative of excellent adsorption efficiency with 1100 mg g^−1^ just in 50 min. Furthermore, it can be regenerated for more than ten cycles by the simple squeezing method due to its unique compressive property. Considering the priority of low-cost, simple process and eco-friendship, the AS adsorbent derived from biomass resources show a bright future of low cost and renewable material resign of MB adsorbents, it suggests that the biomass resource could become a promising candidate for industrially applicable and efficient treatments of dye containing effluents.

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
