# Peer review of "Compressive Alginate Sponge Derived from Seaweed Biomass Resources for Methylene Blue Removal from Wastewater"

_polymers, 2019, doi:10.3390/polym11060961_

Round 1

Reviewer 1 Report

In the present manuscript by Shen et al. the authors report the making of an alginate sponge (AS) using seaweed biomass obtained from Shandong, China. The AS thus prepared is characterized in detail using a suite of characterization techniques. Potential of the AS in use for methylene blue (BM) removal from aqueous solution is investigated in terms of the adsorption capacity and kinetics, mechanical strength, and recyclability. The sponge is found to be capable of removing a good amount of MB in a short time and retains 70% of its MB adsorption capacity even after 10 cycles of use. This is an important original contribution of high significance in the field of aqueous waste-removal. The characterization and investigation of potential applications is thorough. As such this work is definitely suitable for publication in the journal 'polymers'. However, there are some issues (listed below) regarding the presentation of the work that need to be resolved in order to make it easier to read and ensure reproducibility, before publication. 

A major issue with this work is the length of the manuscript. Although there is nothing in the manuscript that can be dispensed with, some parts can be moved to the supplementary information (SI). In general, the paper can be divided into making and characterization of the AS, adsorption capacity and kinetics of the AS, mechanical strength and recyclability of the AS for use in MB removal. While the making and characterization are important for understanding the later sections they do not form the central theme of the paper. Therefore, the figures 1 and 2 and the associated description and discussion can be moved to the SI, while only a few sentences are retained in the main text to note the main conclusions from characterization. This will make the paper shorter and easier to read. Further, dividing the results into subsections as mentioned above will be of help to the reader.

In addition to the above, there are some minor issues like missing of details in the parameters used in some characterizations, e.g. no operating parameters are provided for the SEM. 

There are several typos throughout the manuscript that need to be corrected. For example, line # 162 micrometers misspelt as macrometers, line # 249 value misspelt as vale, line # 119, powder misspelt as power. I would suggest the authors look for possible typos elsewhere and correct them too.

It would be helpful to number the equations used in the text.

Author Response

Reviewer 1:In the present manuscript by Shen et al. the authors report the making of an alginate sponge (AS) using seaweed biomass obtained from Shandong, China. The AS thus prepared is characterized in detail using a suite of characterization techniques. Potential of the AS in use for methylene blue (BM) removal from aqueous solution is investigated in terms of the adsorption capacity and kinetics, mechanical strength, and recyclability. The sponge is found to be capable of removing a good amount of MB in a short time and retains 70% of its MB adsorption capacity even after 10 cycles of use. This is an important original contribution of high significance in the field of aqueous waste-removal. The characterization and investigation of potential applications is thorough. As such this work is definitely suitable for publication in the journal 'polymers'. However, there are some issues (listed below) regarding the presentation of the work that need to be resolved in order to make it easier to read and ensure reproducibility, before publication. 

Comment 1:

A major issue with this work is the length of the manuscript. Although there is nothing in the manuscript that can be dispensed with, some parts can be moved to the supplementary information (SI). In general, the paper can be divided into making and characterization of the AS, adsorption capacity and kinetics of the AS, mechanical strength and recyclability of the AS for use in MB removal. While the making and characterization are important for understanding the later sections they do not form the central theme of the paper. Therefore, the figures 1 and 2 and the associated description and discussioncan be moved to the SI, while only a few sentences are retained in the main text to note the main conclusions from characterization. This will make the paper shorter and easier to read. Further, dividing the results into subsections as mentioned above will be of help to the reader.

Replying: Thanks for reviewer’s comment. According to the reviewer’s recommendation, we have reduced the length of the main parts and also set up subsections of the results (3.1. Material Properties; 3.2. Absorption Properties; 3.3. Compressive properties and recyclability) in revised manuscript. Considering the clearance and integrity for readers to understand, we retained the Figures 1 and 2, while moved the main descriptions and discussions to SI materials.

Revisions:

(1)   3.1. Material Properties 

The as-obtained sponge was elastic enough to be easily handled and cut with desired shapes, such as rods, cylinders, papers or cubes to satisfy various absorption requirements (Fig 1a). Moreover, the AS has ideal density (34.6 mg cm−3) compared with conventional compacted alginate-based materials [54], which could effortlessly stand on a grass leaf (Fig.1b). The strength and lightweight of AS were closely related to its three-dimensional (3D) porous framework with abundant pores as shown in Fig. 1c and d. As a result, the interior sponge possessed a wide range of pore sizes from nanometers to hundreds of micrometers (mostly less than 100 μm) calculated by Mercury Intrusion Porosimetry method (Fig. S1). FT-IR analysis and XRD pattern revealed that the carboxyl groups penetrated in the AS network and AS have none of distinct crystalline formation (Fig. 1e and 1f). Through the photoelectron lines of the wide-scan XPS spectrum (Fig. S2), the presence of carbon (57.72% area), oxygen (39.03% area) and calcium (3.25% area) atoms in AS were confirmed. Further investigation of atomic binding states on the surface of AS was conducted by XPS analysis (Fig. 1g and 1h). (see lines 160 to 173)

(2) Supplementary detailed physical properties of AS and supplementary detailed chemical properties of AS. (see SI Materials)

Comment 2:

In addition to the above, there are some minor issues like missing of details in the parameters used in some characterizations, e.g. no operating parameters are provided for the SEM. 

Replying:Thanks for reviewer’s comment. We have added the missing details of SEM operating parameters in the experiment section. 

Revision: The morphology of the AS was examined by the field-emission scanning electron microscope (SEM) (HitachiS-5500, Japan) operating at 10 kV acceleration voltage. All samples were coated with gold prior to acquiring images.(see lines 122 to 123)

Comment 3:

There are several typos throughout the manuscript that need to be corrected. For example, line # 162 micrometers misspelt as macrometers, line # 249 value misspelt as vale, line # 119, powder misspelt as power. I would suggest the authors look for possible typos elsewhere and correct them too.

Replying:Thanks for reviewer’s comment. These corrects have been revised, furthermore, we carefully check the typos in the whole manuscript.

Comment 4:

It would be helpful to number the equations used in the text. 

Replying: Thanks for reviewer’s comment. We have added the number of the equations in the text.

Revisions: See lines148, 232, 240, 270 and 271.

Reviewer 2 Report

This work reported the fabrication of  alginate sponge (AS) from seaweed biomass resources through a green two-step lyophilization method.

In general the work is good, but there are some comments should be considered before publication as follows:

1. The author should highlight the application environment in the title, such as wastewater, sea water, or what?

2- with regards to the abstract part, the author should provide the merits of the fabricated materials, highlight the objective, and the novelty needs more clarification.

3- With regards to the introduction part:

  (A) It is recommended to highlight different pollutants and their effects on the aquatic environment,
   (B) Further background information on the use of sponge and porous materials in water treatment should be added.
   (C) Different materials used in the same objective.
   (D) benefit and advantage of the current study compared to others

 The following citation may be used:

Chem. Asian J., 10(11), 2467–2478, (2015). ; Arabian Journal of Chemistry, 9(2), S1721–S1728, (2016); ChemPlusChem, 80(7), 1119–1126, (2015)Analyst. 139(24), 6393-6405, (2014)Green Chemistry, 20, 1841-1857, (2018) ChemistrySelect 2 (34), 11083-11090 (2017); Water Resources and Industry 12,  8-24 , (2015). ;  Arabian Journal of Chemistry  10, S3381-S3393(2017). 

   (E) Moreover, the provided claim at the end of the introduction needs more clarification and revision on the basis of the objective of the study, the material and approach, the relevance, the expected, and the key findings.

4- With regards to the synthesis process, it is recommended to add chart that show different steps of preparation. this means scheme (1) should be modified and redesign with more information to help the readers.

5. The first paragraph of the results and discussion should be divided into two or three paragraphs.

6- Figure of the XRD pattern of AS must be modified to show peaks, please adjust the scale of the y axis.

7- Did the author study the possibility of Zeta potential of the prepared material? o

8- Can the author provide the images of the solution before and after treatment?

9- What about the reusability?

10- What about the surface analysis before, and after using?, did the author study the surface analysis after the adsorption process?

11- Conclusion part needs to be modified based on the material preparation, main results obtained, and the future if possible.

Author Response

Reviewer 2:This work reported the fabrication of alginate sponge (AS) from seaweed biomass resources through a green two-step lyophilization method. In general, the work is good, but there are some comments should be considered before publication as follows:

Comment 1:

The author should highlight the application environment in the title, such as wastewater, sea water, or what?

Replying: Thanks for reviewer’s comment. The title of this manuscript has been revised into Compressive alginate sponge derived from seaweed biomass resources for methylene blue removal from wastewater.

Revision:See lines 3 to 4.

Comment 2:

with regards to the abstract part, the author should provide the merits of the fabricated materials, highlight the objective, and the novelty needs more clarification.

Replying:Thanks for reviewer’s comment. We revised the abstract again, which support the merits of the fabricated materials, highlight of the objective and novelty.

Revisions: 

(1) Low cost fabrication of water treatment polymer materials directly from biomass resources is urgently needed in recent days. (see lines 17 to 18)

(2) This material is much different from conventional oven-, air-, vacuum-dried alginate-based adsorbents, which show limitations of shrinkage, rigidness, tight nonporous structure and restricted ions diffusion, hindering its practical applications, and was used to efficiently methylene blue (MB), main colorful contaminant in dye manufacturingfromwastewater.(see lines 19 to 23)

(3) Compressive AS demonstrates its merits of high capability, large efficiency and easy to recycle as well as low cost resources, indicating widespread potentials for application in dye contaminant control regarding environmental protection.(see lines 29 to 32)

Comment 3:

With regards to the introduction part:

(A) It is recommended to highlight different pollutants and their effects on the aquatic environment,

(B) Further background information on the use of sponge and porous materials in water treatment should be added.

(C) Different materials used in the same objective.

(D) benefit and advantage of the current study compared to others

 The following citation may be used:

Chem. Asian J., 10(11), 2467–2478, (2015).; Arabian Journal of Chemistry, 9(2), S1721–S1728, (2016); ChemPlusChem, 80(7), 1119–1126, (2015); Analyst. 139(24), 6393-6405, (2014); Green Chemistry, 20, 1841-1857, (2018); ChemistrySelect 2 (34), 11083-11090 (2017); Water Resources and Industry 12,  8-24 , (2015). ;  Arabian Journal of Chemistry  10, S3381-S3393(2017). 

(E) Moreover, the provided claim at the end of the introduction needs more clarification and revision on the basis of the objective of the study, the material and approach, the relevance, the expected, and the key findings.

Replying: Thanks for reviewer’s comment. We have carefully rewritten and revised introduction part according to the valuable advice.

Revisions: 

(1)  Recently, industrial developments led to the undesirable environmental problems in the world, especially water pollution [1-3]. Many industries like the textile industry produce much wastewater, which contains a number of contaminants, including acidic or caustic dissolved solids, toxic compounds, and any different dyes, many of these dyes are carcinogenic, mutagenic, and teratogenic and also toxic to human beings, fish species, and microorganisms [1,4-8]. Among these contaminants, dyes are common contaminants in water discharged from various industries, such as plastics, textiles, printing, paper and leather industry. Due to its complex aromatic molecular structures, dyes in contaminated water are stable and resistant to biodegradation, which has become a serious environmental problem [9-11]. (see lines 37 to 45)

(2)  Recently, the three-dimensional (3D) porous structures ensure large surface area for efficient dye adsorption, and exhibits the desirable merits of biocompatibility, biodegradability and economic efficiency [30]. Hence, much attention has been paid to developing sponge and porous materials as absorbers and they can achieve oil−water separation via a simple, fast, and effective absorption process [31-35]. Generally, an ideal absorbent material should have high oil absorption capacity, high selectivity, low density, and excellent recyclability, and it should be environmentally friendly. A large variety of low-cost adsorbents including natural species (plant fiber [36], silk cotton [37], leaves [38], etc.), industrial/agricultural wastes or by-product (citrus waste peels [39], bagasse [40], rice husk [41], etc.) and extractive biopolymers (chitosan [42], guar gum [43], alginate [44-47],etc.) have been explored as they are inexpensive, non-toxic and environmentally friendly.(see lines 56 to 65)

(3)  In the paper, we aim at the fabrication of sponge-like adsorbing polymers directly from biomass resources, which is consistent with the object of low cost and renewable material design. Herein, we describe a green two-step lyophilization strategy to prepare a compressive alginate sponge (AS) derived from natural seaweed biomass resources with large surface area, high porosity and controllable morphology. This method has advantages of low cost, simple and eco-friendly approach that does not include toxic or expensive resources, complicated or multistep chemical reactions. In this work, the as-prepared sponge was used to efficiently remove methylene blue (MB), main colorful contaminant in dye manufacturing, from aqueous solution through batch and continuous fixed-bed column adsorption. Their kinetics, thermodynamic analysis and adsorption mechanism are investigated. More importantly, the regeneration of the compressive sponge just by squeezing demonstrates its large potential to be used in practical applications for water treatment.(see lines 77 to 87)

Comment 4:

With regards to the synthesis process, it is recommended to add chart that show different steps of preparation. this means scheme (1) should be modified and redesign with more information to help the readers.

Replying: Thanks for reviewer’s comment. In combination of all the reviewer’s comments of preparation steps. We have made several main revisions to make more clear information for readers: we have moved the Scheme 1 to the synthesis parts of the revised manuscript with detailed descriptions of preparation steps for the sponge materials; we have moved the supplementary synthesis part to SI Materials; we have added the involved chemical equations of Scheme 1 in SI Materials. 

Revisions: 

(1)   2.3. Preparation of AS. The AS was prepared by a green two-step lyophilization method from seaweed biomass resources which was presented in Scheme 1.(see lines 104 to 106)

(2)   Typically, sodium alginate was extracted from seaweed according to our previous work [43] and it was highly soluble in water to obtain aqueous solution. After the first-step-lyophilization at −50 oC for 48 h, a freestanding while fragile lyophilized sample was obtained initially. Once thoroughly immersed the lyophilized sample in 5 wt % CaCl2aqueous solution for 12 h, the Ca2+ions were diffused and penetrated into the network of the lyophilized sample through ionic reaction. After then, the second-step-lyophilization was processed to obtain a final AS.(see SI Materials)

(3)   For clear understanding, the involved chemical equations of different steps in Scheme 1 was supported in Scheme S1.(see lines 115 to 116 and Scheme S1)

Comment 5:

The first paragraph of the results and discussion should be divided into two or three paragraphs.

Replying: Thanks for reviewer’s comment. In combination of all the reviewers’ comments, we have divided and shortened the first paragraph of the results and discussion, and moved descriptions and discussions into SI Materials. In order to make readers more understandable, we also set up the subsections of the results and discussions in the revised manuscript.

Revisions: 

(1) 3.1. Material Properties. The as-obtained sponge was elastic enough to be easily handled and cut with desired shapes, such as rods, cylinders, papers or cubes to satisfy various absorption requirements (Fig 1a). Moreover, the AS has ideal density (34.6 mg cm−3) compared with conventional compacted alginate-based materials [54], which could effortlessly stand on a grass leaf (Fig.1b). The strength and lightweight of AS were closely related to its three-dimensional (3D) porous framework with abundant pores as shown in Fig. 1c and d. As a result, the interior sponge possessed a wide range of pore sizes from nanometers to hundreds of micrometers (mostly less than 100 μm) calculated by Mercury Intrusion Porosimetry method (Fig. S1). FT-IR analysis and XRD pattern revealed that the carboxyl groups penetrated in the AS network and AS have none of distinct crystalline formation (Fig. 1e and 1f). Through the photoelectron lines of the wide-scan XPS spectrum (Fig. S2), the presence of carbon (57.72% area), oxygen (39.03% area) and calcium (3.25% area) atoms in AS were confirmed. Further investigation of atomic binding states on the surface of AS was conducted by XPS analysis (Fig. 1g and 1h).(see lines 160 to 173)

(2) Supplementary detailed physical properties of AS and supplementary detailed chemical properties of AS. (see SI Materials)

Comment 6:

Figure of the XRD pattern of AS must be modified to show peaks, please adjust the scale of the y axis.

ReplyingThanks for reviewer’s comment. We have modified the Figure of the XRD pattern of AS in the revised manuscript according to your kind suggestion.

Revision: See line 174, Fig. 1f.

Comment 7:

Did the author study the possibility of Zeta potential of the prepared material? 

Replying: Thanks for reviewer’s comment. We have added the Zeta potential data in the revised manuscript.

Revision: However, when the pH value was higher than 4.0 (above the pKa value of MB), the surface charges of AS became more negative which was confirmed by Zeta potential data, promoting the electrostatic interaction between surface active sites and adsorbate, improving the adsorption capacity for MB. (see lines 193 to 196)

Comment 8:

Can the author provide the images of the solution before and after treatment?

Replying: Thanks for reviewer’s comment. We have added the images of the solution before and after treatment.

Revision: A clear comparison of removal effect could be presented by the images before and after batch adsorption (Fig. S3).(see lines 216 to 217 and Fig. S3)

Comment 9:

What about the reusability?

Replying: We thank reviewer’s comment very much. We have demonstrated the reusability of AS for the removal of MB as reviewer’s suggestion. In the main text, we wrote that “As shown in Fig. 5c, the adsorption capacity for MB was maintained for about 70% after 10 cycles with 861 mg g−1, exhibiting good recyclability upon easy squeezing method. The main reason for the residual MB during recycled use may be attributed to the stability of partial trapped MB molecules complexed within the network of AS chains. For further thoroughly removal of MB during the cyclic tests, chemical cleaning or heat treatment might be efficient.” (see lines 355 to 360)

Comment 10:

What about the surface analysis before, and after using? did the author study the surface analysis after the adsorption process?

Replying: Thanks for reviewer’s comment. Considering the main objective of the manuscript is focused on the design of sponge-like adsorbents for MB removal, we may concern more about the internal 3D multiporous structure instead of surface structure as particle adsorbents. Thus, according to your suggestion, we supplemented the FT-IR spectra before and after MB adsorption to study the structural analysis after the adsorption processwhich we expect maybe more convincing for our designed materials.

Revision: In additional, the AS before and after adsorbing methylene blue were characterized by FT-IR spectra. As shown in Fig. S4, the infrared characteristic peaks (1607 and 1422 cm−1) of the C=O group shifted to low Infrared absorption peak, which suggested that internal porous structure of AS was favorable to the electrostatic interaction between the polymer carboxylic functional groups and cationic groups of the MB.(see lines 217 to 222)

Comment 11:

Conclusion part needs to be modified based on the material preparation, main results obtained, and the future if possible.

Replying: Thanks for reviewer’s comment. According to your important advice, we revised the conclusion part based on the material preparation, main results obtained, and the future if possible.

Revision: A compressive AS was prepared from seaweed biomass resources via a green two-step lyophilization method, which was used to remove MB from wastewater for pollutant control. Due to the 3D multiporous structure, large specific surface area and sufficient active sites, as-prepared AS exhibited high adsorption capacity (1279 mg g−1) for MB removal, which is much different from the conventional oven-, air-, vacuum-dried alginate-based adsorbents, which show limitations of shrinkage, rigidness, tight nonporous structure and restricted ions diffusion, hindering its practical applications. Their kinetics, thermodynamic analysis and adsorption mechanism are investigated. Importantly, the freestanding and elastic AS was facilitated to assemble into a column-packed device for fixed-bed continuous wastewater treatment, indicative excellent adsorption efficiency with 1100 mg g−1 just in 50 min. Furthermore, it can be regenerated for more than ten cycles by simple squeezing method due to its unique compressive property. Considering the priority of low-cost, simple process and eco-friendship, the AS adsorbent derived from biomass resources show a bright future of low cost and renewable material resign of MB adsorbents, it suggests that biomass resource could become a promising candidate for industrially applicable and efficient treatments of dye containing effluents.(see lines 370 to 383)

Reviewer 3 Report

After reading the manuscript entitled: Compressive alginate sponge derived from seaweed biomass resources for methylene blue removal, I found the idea and application very interested, however, the manuscript is not suitable to be published in the current shape due to the following reasons:

1- Ithenticate software for detecting plagiarism showed 51% with references and 40% without, and this is considered very high and should not exceed 10%.

2- In page 1, line 22, a bed-fixed absorption should be fixed-bed adsorption.

3- The synthesis of the adsorbent includes successive steps with the addition of chemicals each step. I would be more useful if the reactions described in words in the synthesis part be written as chemical equations.

4- In page 3, line 103, BET analysis was mentioned, but it was not discussed in the results.

5- Usually for any developed adsorbent to compete with other adsorbents it should have high sensitivity at low concentration, while in this work all used concentration are considered very high. It is important to run experiments at lower concentrations such as 10 and 5 ppm.

6- In page 4, line 137, the authors mentioned pumping the solution through the packed be but did not mention the flow rate used, it is important to clearly mention it as it has an important effect on the performance.

7- The particle size and particle size distribution of the adsorbent packed in the column should be mentioned as well as the mass of adsorbent used each time to fill the column.

8-In page 4, lines 139-149 are more related to experimental work than to discussion of results.

9- In page 4, lines 155-156, this statement is not clear, why did you use ice? This was not mentioned in the experimental part.

10-Figure 1b is not clear, either color the column or use a colored background.

11- In page 6, line 190, it was mentioned that XPS in figure S2 while it is shown in Figure 2C and 2D. 

12- In figure 2a, it would be more useful to show FTIR for the weed extract before any treatment, after each step, and after adsorption of MB to confirm the proposed steps of preparation and occurrence of adsorption on the surface.

13- It is very useful to measure the point of zero charges of the adsorbent to confirm the discussed pH effect. 

14- In page 12, lines 353-354, the dimension of the column seem not correct, please check.

15- I suggest that all characterization figures be in the main body of the manuscript and not in supplementary data.

16- English editing is required.

Author Response

Point-by-Point Responses to the Comments of Reviewers

Reviewer 1:In the present manuscript by Shen et al. the authors report the making of an alginate sponge (AS) using seaweed biomass obtained from Shandong, China. The AS thus prepared is characterized in detail using a suite of characterization techniques. Potential of the AS in use for methylene blue (BM) removal from aqueous solution is investigated in terms of the adsorption capacity and kinetics, mechanical strength, and recyclability. The sponge is found to be capable of removing a good amount of MB in a short time and retains 70% of its MB adsorption capacity even after 10 cycles of use. This is an important original contribution of high significance in the field of aqueous waste-removal. The characterization and investigation of potential applications is thorough. As such this work is definitely suitable for publication in the journal 'polymers'. However, there are some issues (listed below) regarding the presentation of the work that need to be resolved in order to make it easier to read and ensure reproducibility, before publication. 

Comment 1:

A major issue with this work is the length of the manuscript. Although there is nothing in the manuscript that can be dispensed with, some parts can be moved to the supplementary information (SI). In general, the paper can be divided into making and characterization of the AS, adsorption capacity and kinetics of the AS, mechanical strength and recyclability of the AS for use in MB removal. While the making and characterization are important for understanding the later sections they do not form the central theme of the paper. Therefore, the figures 1 and 2 and the associated description and discussion can be moved to the SI, while only a few sentences are retained in the main text to note the main conclusions from characterization. This will make the paper shorter and easier to read. Further, dividing the results into subsections as mentioned above will be of help to the reader.

Replying: Thanks for reviewer’s comment. According to the reviewer’s recommendation, we have reduced the length of the main parts and also set up subsections of the results (3.1. Material Properties; 3.2. Absorption Properties; 3.3. Compressive properties and recyclability) in revised manuscript. Considering the clearance and integrity for readers to understand, we retained the Figures 1 and 2, while moved the main descriptions and discussions to SI materials.

Revisions:

(1)   3.1. Material Properties 

The as-obtained sponge was elastic enough to be easily handled and cut with desired shapes, such as rods, cylinders, papers or cubes to satisfy various absorption requirements (Fig 1a). Moreover, the AS has ideal density (34.6 mg cm−3) compared with conventional compacted alginate-based materials [54], which could effortlessly stand on a grass leaf (Fig.1b). The strength and lightweight of AS were closely related to its three-dimensional (3D) porous framework with abundant pores as shown in Fig. 1c and d. As a result, the interior sponge possessed a wide range of pore sizes from nanometers to hundreds of micrometers (mostly less than 100 μm) calculated by Mercury Intrusion Porosimetry method (Fig. S1). FT-IR analysis and XRD pattern revealed that the carboxyl groups penetrated in the AS network and AS have none of distinct crystalline formation (Fig. 1e and 1f). Through the photoelectron lines of the wide-scan XPS spectrum (Fig. S2), the presence of carbon (57.72% area), oxygen (39.03% area) and calcium (3.25% area) atoms in AS were confirmed. Further investigation of atomic binding states on the surface of AS was conducted by XPS analysis (Fig. 1g and 1h). (see lines 160 to 173)

(2) Supplementary detailed physical properties of AS and supplementary detailed chemical properties of AS. (see SI Materials)

Comment 2:

In addition to the above, there are some minor issues like missing of details in the parameters used in some characterizations, e.g. no operating parameters are provided for the SEM. 

Replying:Thanks for reviewer’s comment. We have added the missing details of SEM operating parameters in the experiment section. 

Revision: The morphology of the AS was examined by the field-emission scanning electron microscope (SEM) (HitachiS-5500, Japan) operating at 10 kV acceleration voltage. All samples were coated with gold prior to acquiring images.(see lines 122 to 123)

Comment 3:

There are several typos throughout the manuscript that need to be corrected. For example, line # 162 micrometers misspelt as macrometers, line # 249 value misspelt as vale, line # 119, powder misspelt as power. I would suggest the authors look for possible typos elsewhere and correct them too.

Replying:Thanks for reviewer’s comment. These corrects have been revised, furthermore, we carefully check the typos in the whole manuscript.

Comment 4:

It would be helpful to number the equations used in the text. 

Replying: Thanks for reviewer’s comment. We have added the number of the equations in the text.

Revisions: See lines148, 232, 240, 270 and 271.

Reviewer 2:This work reported the fabrication of alginate sponge (AS) from seaweed biomass resources through a green two-step lyophilization method. In general, the work is good, but there are some comments should be considered before publication as follows:

Comment 1:

The author should highlight the application environment in the title, such as wastewater, sea water, or what?

Replying: Thanks for reviewer’s comment. The title of this manuscript has been revised into Compressive alginate sponge derived from seaweed biomass resources for methylene blue removal from wastewater.

Revision:See lines 3 to 4.

Comment 2:

with regards to the abstract part, the author should provide the merits of the fabricated materials, highlight the objective, and the novelty needs more clarification.

Replying:Thanks for reviewer’s comment. We revised the abstract again, which support the merits of the fabricated materials, highlight of the objective and novelty.

Revisions: 

(1) Low cost fabrication of water treatment polymer materials directly from biomass resources is urgently needed in recent days. (see lines 17 to 18)

(2) This material is much different from conventional oven-, air-, vacuum-dried alginate-based adsorbents, which show limitations of shrinkage, rigidness, tight nonporous structure and restricted ions diffusion, hindering its practical applications, and was used to efficiently methylene blue (MB), main colorful contaminant in dye manufacturingfromwastewater.(see lines 19 to 23)

(3) Compressive AS demonstrates its merits of high capability, large efficiency and easy to recycle as well as low cost resources, indicating widespread potentials for application in dye contaminant control regarding environmental protection.(see lines 29 to 32)

Comment 3:

With regards to the introduction part:

(A) It is recommended to highlight different pollutants and their effects on the aquatic environment,

(B) Further background information on the use of sponge and porous materials in water treatment should be added.

(C) Different materials used in the same objective.

(D) benefit and advantage of the current study compared to others

 The following citation may be used:

Chem. Asian J., 10(11), 2467–2478, (2015).; Arabian Journal of Chemistry, 9(2), S1721–S1728, (2016); ChemPlusChem, 80(7), 1119–1126, (2015); Analyst. 139(24), 6393-6405, (2014); Green Chemistry, 20, 1841-1857, (2018); ChemistrySelect 2 (34), 11083-11090 (2017); Water Resources and Industry 12,  8-24 , (2015). ;  Arabian Journal of Chemistry  10, S3381-S3393(2017). 

(E) Moreover, the provided claim at the end of the introduction needs more clarification and revision on the basis of the objective of the study, the material and approach, the relevance, the expected, and the key findings.

Replying: Thanks for reviewer’s comment. We have carefully rewritten and revised introduction part according to the valuable advice.

Revisions: 

(1)  Recently, industrial developments led to the undesirable environmental problems in the world, especially water pollution [1-3]. Many industries like the textile industry produce much wastewater, which contains a number of contaminants, including acidic or caustic dissolved solids, toxic compounds, and any different dyes, many of these dyes are carcinogenic, mutagenic, and teratogenic and also toxic to human beings, fish species, and microorganisms [1,4-8]. Among these contaminants, dyes are common contaminants in water discharged from various industries, such as plastics, textiles, printing, paper and leather industry. Due to its complex aromatic molecular structures, dyes in contaminated water are stable and resistant to biodegradation, which has become a serious environmental problem [9-11]. (see lines 37 to 45)

(2)  Recently, the three-dimensional (3D) porous structures ensure large surface area for efficient dye adsorption, and exhibits the desirable merits of biocompatibility, biodegradability and economic efficiency [30]. Hence, much attention has been paid to developing sponge and porous materials as absorbers and they can achieve oil−water separation via a simple, fast, and effective absorption process [31-35]. Generally, an ideal absorbent material should have high oil absorption capacity, high selectivity, low density, and excellent recyclability, and it should be environmentally friendly. A large variety of low-cost adsorbents including natural species (plant fiber [36], silk cotton [37], leaves [38], etc.), industrial/agricultural wastes or by-product (citrus waste peels [39], bagasse [40], rice husk [41], etc.) and extractive biopolymers (chitosan [42], guar gum [43], alginate [44-47],etc.) have been explored as they are inexpensive, non-toxic and environmentally friendly.(see lines 56 to 65)

(3)  In the paper, we aim at the fabrication of sponge-like adsorbing polymers directly from biomass resources, which is consistent with the object of low cost and renewable material design. Herein, we describe a green two-step lyophilization strategy to prepare a compressive alginate sponge (AS) derived from natural seaweed biomass resources with large surface area, high porosity and controllable morphology. This method has advantages of low cost, simple and eco-friendly approach that does not include toxic or expensive resources, complicated or multistep chemical reactions. In this work, the as-prepared sponge was used to efficiently remove methylene blue (MB), main colorful contaminant in dye manufacturing, from aqueous solution through batch and continuous fixed-bed column adsorption. Their kinetics, thermodynamic analysis and adsorption mechanism are investigated. More importantly, the regeneration of the compressive sponge just by squeezing demonstrates its large potential to be used in practical applications for water treatment.(see lines 77 to 87)

Comment 4:

With regards to the synthesis process, it is recommended to add chart that show different steps of preparation. this means scheme (1) should be modified and redesign with more information to help the readers.

Replying: Thanks for reviewer’s comment. In combination of all the reviewer’s comments of preparation steps. We have made several main revisions to make more clear information for readers: we have moved the Scheme 1 to the synthesis parts of the revised manuscript with detailed descriptions of preparation steps for the sponge materials; we have moved the supplementary synthesis part to SI Materials; we have added the involved chemical equations of Scheme 1 in SI Materials. 

Revisions: 

(1)   2.3. Preparation of AS. The AS was prepared by a green two-step lyophilization method from seaweed biomass resources which was presented in Scheme 1.(see lines 104 to 106)

(2)   Typically, sodium alginate was extracted from seaweed according to our previous work [43] and it was highly soluble in water to obtain aqueous solution. After the first-step-lyophilization at −50 oC for 48 h, a freestanding while fragile lyophilized sample was obtained initially. Once thoroughly immersed the lyophilized sample in 5 wt % CaCl2aqueous solution for 12 h, the Ca2+ions were diffused and penetrated into the network of the lyophilized sample through ionic reaction. After then, the second-step-lyophilization was processed to obtain a final AS.(see SI Materials)

(3)   For clear understanding, the involved chemical equations of different steps in Scheme 1 was supported in Scheme S1.(see lines 115 to 116 and Scheme S1)

Comment 5:

The first paragraph of the results and discussion should be divided into two or three paragraphs.

Replying: Thanks for reviewer’s comment. In combination of all the reviewers’ comments, we have divided and shortened the first paragraph of the results and discussion, and moved descriptions and discussions into SI Materials. In order to make readers more understandable, we also set up the subsections of the results and discussions in the revised manuscript.

Revisions: 

(1) 3.1. Material Properties. The as-obtained sponge was elastic enough to be easily handled and cut with desired shapes, such as rods, cylinders, papers or cubes to satisfy various absorption requirements (Fig 1a). Moreover, the AS has ideal density (34.6 mg cm−3) compared with conventional compacted alginate-based materials [54], which could effortlessly stand on a grass leaf (Fig.1b). The strength and lightweight of AS were closely related to its three-dimensional (3D) porous framework with abundant pores as shown in Fig. 1c and d. As a result, the interior sponge possessed a wide range of pore sizes from nanometers to hundreds of micrometers (mostly less than 100 μm) calculated by Mercury Intrusion Porosimetry method (Fig. S1). FT-IR analysis and XRD pattern revealed that the carboxyl groups penetrated in the AS network and AS have none of distinct crystalline formation (Fig. 1e and 1f). Through the photoelectron lines of the wide-scan XPS spectrum (Fig. S2), the presence of carbon (57.72% area), oxygen (39.03% area) and calcium (3.25% area) atoms in AS were confirmed. Further investigation of atomic binding states on the surface of AS was conducted by XPS analysis (Fig. 1g and 1h).(see lines 160 to 173)

(2) Supplementary detailed physical properties of AS and supplementary detailed chemical properties of AS. (see SI Materials)

Comment 6:

Figure of the XRD pattern of AS must be modified to show peaks, please adjust the scale of the y axis.

ReplyingThanks for reviewer’s comment. We have modified the Figure of the XRD pattern of AS in the revised manuscript according to your kind suggestion.

Revision: See line 174, Fig. 1f.

Comment 7:

Did the author study the possibility of Zeta potential of the prepared material? 

Replying: Thanks for reviewer’s comment. We have added the Zeta potential data in the revised manuscript.

Revision: However, when the pH value was higher than 4.0 (above the pKa value of MB), the surface charges of AS became more negative which was confirmed by Zeta potential data, promoting the electrostatic interaction between surface active sites and adsorbate, improving the adsorption capacity for MB. (see lines 193 to 196)

Comment 8:

Can the author provide the images of the solution before and after treatment?

Replying: Thanks for reviewer’s comment. We have added the images of the solution before and after treatment.

Revision: A clear comparison of removal effect could be presented by the images before and after batch adsorption (Fig. S3).(see lines 216 to 217 and Fig. S3)

Comment 9:

What about the reusability?

Replying: We thank reviewer’s comment very much. We have demonstrated the reusability of AS for the removal of MB as reviewer’s suggestion. In the main text, we wrote that “As shown in Fig. 5c, the adsorption capacity for MB was maintained for about 70% after 10 cycles with 861 mg g−1, exhibiting good recyclability upon easy squeezing method. The main reason for the residual MB during recycled use may be attributed to the stability of partial trapped MB molecules complexed within the network of AS chains. For further thoroughly removal of MB during the cyclic tests, chemical cleaning or heat treatment might be efficient.” (see lines 355 to 360)

Comment 10:

What about the surface analysis before, and after using? did the author study the surface analysis after the adsorption process?

Replying: Thanks for reviewer’s comment. Considering the main objective of the manuscript is focused on the design of sponge-like adsorbents for MB removal, we may concern more about the internal 3D multiporous structure instead of surface structure as particle adsorbents. Thus, according to your suggestion, we supplemented the FT-IR spectra before and after MB adsorption to study the structural analysis after the adsorption processwhich we expect maybe more convincing for our designed materials.

Revision: In additional, the AS before and after adsorbing methylene blue were characterized by FT-IR spectra. As shown in Fig. S4, the infrared characteristic peaks (1607 and 1422 cm−1) of the C=O group shifted to low Infrared absorption peak, which suggested that internal porous structure of AS was favorable to the electrostatic interaction between the polymer carboxylic functional groups and cationic groups of the MB.(see lines 217 to 222)

Comment 11:

Conclusion part needs to be modified based on the material preparation, main results obtained, and the future if possible.

Replying: Thanks for reviewer’s comment. According to your important advice, we revised the conclusion part based on the material preparation, main results obtained, and the future if possible.

Revision: A compressive AS was prepared from seaweed biomass resources viaa green two-step lyophilization method, which was used to remove MB from wastewater for pollutant control. Due to the 3D multiporous structure, large specific surface area and sufficient active sites, as-prepared AS exhibited high adsorption capacity (1279 mg g−1) for MB removal, which is much different from the conventional oven-, air-, vacuum-dried alginate-based adsorbents, which show limitations of shrinkage, rigidness, tight nonporous structure and restricted ions diffusion, hindering its practical applications. Their kinetics, thermodynamic analysis and adsorption mechanism are investigated. Importantly, the freestanding and elastic AS was facilitated to assemble into a column-packed device for fixed-bed continuous wastewater treatment, indicative excellent adsorption efficiency with 1100 mg g−1 just in 50 min. Furthermore, it can be regenerated for more than ten cycles by simple squeezing method due to its unique compressive property. Considering the priority of low-cost, simple process and eco-friendship, the AS adsorbent derived from biomass resources show a bright future of low cost and renewable material resign of MB adsorbents, it suggests that biomass resource could become a promising candidate for industrially applicable and efficient treatments of dye containing effluents.(see lines 370 to 383)

Reviewer 3:After reading the manuscript entitled: Compressive alginate sponge derived from seaweed biomass resources for methylene blue removal, I found the idea and application very interested, however, the manuscript is not suitable to be published in the current shape due to the following reasons:

Comment 1:

Ithenticate software for detecting plagiarism showed 51% with references and 40% without, and this is considered very high and should not exceed 10%.

Replying: Thanks for reviewer’s comment, we have carefully checked and modified our revised manuscript for detecting plagiarism according to reviewer’s suggest and we promise that this manuscript is originally produced by our authors.

Comment 2:

In page 1, line 22, a bed-fixed absorption should be fixed-bed adsorption.

Replying:Thanks for reviewer’s comment.bed-fixed” has revised into fixed-bed” in the manuscript. 

Revision: See lines 26, 84, 152, 334, 342, 348, 355 and 378.

Comment 3:

The synthesis of the adsorbent includes successive steps with the addition of chemicals each step. I would be more useful if the reactions described in words in the synthesis part be written as chemical equations.

Replying: Thanks for reviewer’s comment.We added the involved chemical equations of successive steps with the addition of chemicals. 

Revision: For clear understanding, he involved chemical equations of different steps in Scheme 1 was supported in Scheme S1.(see lines 115 to 116 and Scheme S1)

Comment 4:

In page 3, line 103, BET analysis was mentioned, but it was not discussed in the results.

Replying:Thanks for reviewer’s comment.In fact, we have tested BET analysis, while the sponge is too light to obtain an accurate and stable BET data for several times, instead we use the Mercury Intrusion Porosimetry method to characterize the pore sizes of sponge. According to reviewer’s suggest, we have deleted the characterization method of BET analysis for clear reading.

Comment 5:

Usually for any developed adsorbent to compete with other adsorbents it should have high sensitivity at low concentration, while in this work all used concentration are considered very high. It is important to run experiments at lower concentrations such as 10 and 5 ppm.

Replying: Thanks for reviewer’s comment. We have supplemented the adsorption experiments at low concentrations (5 mg/L and 10 mg/L) and added the description of the sensitivity in the revised manuscript according to reviewer’s advice.

Revision: Even at low concentration of 5 and 10 mg L−1, the AS also has high sensitivity for efficient MB removal, which is important for adsorbents used in water treatment.(see lines 214 to 216 and Fig. 2b)

Comment 6:

In page 4, line 137, the authors mentioned pumping the solution through the packed be but did not mention the flow rate used, it is important to clearly mention it as it has an important effect on the performance.

Replying:Thanks for reviewer’s comment.Indeed, the flow rate of fixed-bed column adsorption is much important for the adsorption performance. The flow rate of fixed-bed column adsorption has been added in the revised manuscript.

Revision: 50 mL MB solution with an initial concentration of 1000 mg L−1was pumped through the column in a down-flow mode using a peristaltic pumpwith the rate of 10 ml min−1at room temperature.(see lines 156 to 158)

Comment 7:

The particle size and particle size distribution of the adsorbent packed in the column should be mentioned as well as the mass of adsorbent used each time to fill the column.

Replying:Thanks for reviewer’s comment.The diameter, height and weight of adsorbent used each time to fill the column has been added in the revised manuscript. 

Revision: AS were cut with desired column(diameter: 0.8 cm, height: 0.45 cm, weight: 7.8 mg) and then packed into the glass column tightly.(see lines 153 to 155)

Comment 8:

In page 4, lines 139-149 are more related to experimental work than to discussion of results.

Replying: Thanks for reviewer’s comment.Considering all the reviewer’s comments, we have moved the related parts to SI Materials for clear understanding (see page 1, SI Materials).

Comment 9:

In page 4, lines 155-156, this statement is not clear, why did you use ice? This was not mentioned in the experimental part.

Replying:Thanks for reviewer’s comment. The sponge was prepared by a two-step lyophilization processes. In these processes, the alginate solution or lyophilized sodium alginate sample rinsed with 5 wt% CaCl2aqueous solution was firstly frozen, the water in this solution would become ice. In order to make it to be easily understandable, we revised the descriptions in the revised manuscript. 

Revision: The formation of multi-porous structures of AS was mainly induced by the template of ice from solutions under lyphilization. During thetwo-step lyophilization processes, water from solutions were rapidly freezing and then transformed into vapor without passing through the liquid phase under high vacuum condition.(see page 1, SI Materials)

Comment 10:

Figure 1b is not clear, either color the column or use a colored background.

Replying: Thanks for reviewer’s comment. The Fig. 1b has been replaced for a more clear photograph.

Revision: See Fig. 1b.

Comment 11:

In page 6, line 190, it was mentioned that XPS in figure S2 while it is shown in Figure 2C and 2D. 

Replying: Thanks for reviewer’s comment. We have previously mentioned that Fig. S2 was the photoelectron lines of the wide-scan XPS spectrum, and Fig. 2g and h (original Fig. 2C and 2D) was atomic binding states on the surface of AS conducted by XPS analysis.

Comment 12:

In figure 2a, it would be more useful to show FTIR for the weed extract before any treatment, after each step, and after adsorption of MB to confirm the proposed steps of preparation and occurrence of adsorption on the surface.

Replying: Thanks for reviewer’s comment.The FT-IR spectra of sodium alginate, AS and MB adsorbed onto AShas been added according to your valuable suggestion. 

Revision: In additional, the AS before and after adsorbing methylene blue were characterized by FT-IR spectra. As shown in Fig. S4, the infrared characteristic peaks (1607 and 1422 cm−1) of the C=O group shifted to low Infrared absorption peak, which suggested that internal porous structure of AS was favorable to the electrostatic interaction between the polymer carboxylic functional groups and cationic groups of the MB.(see lines 217 to 222 and Fig. S4)

Comment 13:

It is very useful to measure the point of zero charges of the adsorbent to confirm the discussed pH effect. 

Replying: Thanks for reviewer’s comment. We have added the Zeta potential data to confirm the discussed pH effect in the revised manuscript.

Revision: However, when the pH value was higher than 4.0 (above the pKa value of MB), the surface charges of AS became more negative which was confirmed by Zeta potential data, promoting the electrostatic interaction between surface active sites and adsorbate, improving the adsorption capacity for MB. (see lines 193 to 196)

Comment 14:

In page 12, lines 353-354, the dimension of the column seem not correct, please check.

Replying: Thanks for reviewer’s comment. The dimension of the column is the sponge samples rather than the adsorption column device, we have modified the description to avoid confusion.

Revision: Thus, we fabricated a free-standing AS sample (height 1.2 cm, volume capacity 15 ml, internal diameter 0.8 cm) into a flexible column devie for fast removing MB from water (Fig. 4a). (see lines 332 to 334)

Comment 15:

I suggest that all characterization figures be in the main body of the manuscript and not in supplementary data.

Replying: Thanks for reviewer’s comment. In combination of all the reviewer’s comments, we have modified the orders of the characterization figures and emphasized the main three parts in results and discussions: 3.1. Material Properties; 3.2. Absorption Properties; 3.3. Compressive properties and recyclability. We hope it could work for the readers to understand the key point of our work.

Comment 16:

English editing is required.

Replying: Thanks for reviewer’s comment. We have carefully corrected the spelling mistake and grammar problem in the manuscript and we also invited English speaking natives to help for the English editing.

Round 2

Reviewer 3 Report

Accept